# Domain adaptable language modeling of chemical compounds identifies potent pathoblockers for *Pseudomonas aeruginosa*

Georgios Kallergis[1,2], Ehsannedin Asgari[1,9], Martin Empting [3,4,5], Anna K. H. Hirsch [4,5,6], Frank Klawonn[7,8] & Alice C. McHardy [1,2,4] ✉

Computational techniques for predicting molecular properties are emerging as key components for streamlining drug development, optimizing time and financial investments. Here, we introduce ChemLM, a transformer language model for this task. ChemLM leverages self-supervised domain adaptation on chemical molecules to enhance its predictive performance. Within the framework of ChemLM, chemical compounds are conceptualized as sentences composed of distinct chemical 'words', which are employed for training a specialized chemical language model. On the standard benchmark datasets, ChemLM either matched or surpassed the performance of current state-of-the-art methods. Furthermore, we evaluated the effectiveness of ChemLM in identifying highly potent pathoblockers targeting *Pseudomonas aeruginosa* (PA), a pathogen that has shown an increased prevalence of multidrug-resistant strains and has been identified as a critical priority for the development of new medications. ChemLM demonstrated substantially higher accuracy in identifying highly potent pathoblockers against PA when compared to state-of-the-art approaches. An intrinsic evaluation demonstrated the consistency of the chemical language model's representation concerning chemical properties. The results from benchmarking, experimental data and intrinsic analysis of the ChemLM space confirm the wide applicability of ChemLM for enhancing molecular property prediction within the chemical domain.

Approximately 12 years[1] and 1.8$ billion are typically required before a drug reaches the market[2] and there is an overall failure rate of 96% for candidate compounds[3]. The discovery and development of novel anti-infectives, especially against bacterial pathogens are challenging and prone to setbacks[4]. Despite unmet medical needs and the steadily increasing threat of antimicrobial resistance (AMR), the lack of new antibiotics with novel, resistance-breaking modes of action has resulted in an 'innovation gap', potentially leading to a 'post-antibiotic era'[5]. In this scenario, the available treatment options for bacterial infections become ineffective, primarily due to the spread of multi- and pan-resistant strains. This is already evident with pathogens like *Pseudomonas aeruginosa*, frequently found with multiple drug resistances in clinical settings[6]. Consequently, the World Health Organization (WHO) has identified the need for new antibiotics targeting this bacterium as a critical priority.

Languages consist of sequences from finite elements[7], making the distributional hypothesis applicable: "A word is characterized by the

[1]Computational Biology of Infection Research, Helmholtz Centre for Infection Research, Braunschweig, Germany. [2]Braunschweig Integrated Centre of Systems Biology (BRICS), Technische Universität Braunschweig, Braunschweig, Germany. [3]Antiviral & Antivirulence Drugs (AVID), Helmholtz-Institute for Pharmaceutical Research Saarland (HIPS)-Helmholtz Centre for Infection Research (HZI), Saarbrücken, Germany. [4]Deutsches Zentrum für Infektionsforschung (DZIF), Hannover-Braunschweig, Germany. [5]Department of Pharmacy, Saarland University, Campus E8.1, 66123 Saarbrücken, Germany. [6]Department of Drug Design and Optimization (DDOP), Helmholtz-Institute for Pharmaceutical Research Saarland (HIPS)-Helmholtz Centre for Infection Research (HZI), Saarbrücken, Germany. [7]Biostatistics Research Group, Helmholtz Centre for Infection Research, Braunschweig, Germany. [8]Institute for Information Engineering, Ostfalia University of Applied Sciences, 38302 Wolfenbüttel, Germany. [9]Present address: Qatar Computing Research Institute (QCRI), Doha, Qatar. ✉e-mail: alice.mchardy@helmholtz-hzi.de

company it keeps"[8]. Thus, language processing techniques leverage contextual similarities[9,10], aiding applications in protein, DNA, and chemical sequences[11–16]. SMILES, which stands for Simplified Molecular-Input Line-Entry System[17] aligns with this linguistic framework[18], enabling its use in language models such as Word2Vec[9,19,20], RNNs[18,21] in the past, and, more recently, Transformers[22]. Transformers leverage large chemical sequence datasets through transfer learning[23,24]. This approach pretrains models on broad tasks with abundant data before fine-tuning them for specific applications, enhancing performance and convergence speed. Initially developed for supervised learning, transfer learning now extends to self-supervised tasks[25–27], allowing model pretraining on massive datasets.

Here, we describe ChemLM, a language model for efficient transfer learning for chemical compounds. ChemLM utilizes the SMILES representation of molecules as sentences of the input language and a three-stage training process for predicting a specific molecular property of chemical compounds. This includes pretraining of a self-supervised language model on large datasets, self-supervised training on further domain-specific data and subsequent model optimization in a supervised setting. With this, we aimed for a model that can be applied for real-world datasets of experimental compounds that comprise of limited training samples/compounds. We assessed whether language models' training using domain adaptation, which allows us to adapt the pre-trained model on further data from the target domain, enhances the model's predictive ability. We performed extensive performance comparisons to the state-of-the-art models. We furthermore investigated whether the model successfully captures the underlying chemical information and reproduces the chemical space. Moreover, we predicted the potency of candidate pathoblocker compounds against *Pseudomonas aeruginosa* from an experimental dataset encompassing just 219 compounds, demonstrating the value of ChemLM for this application in the drug discovery process.

## Results

### The ChemLM method

ChemLM is a transformer-based model that processes molecular SMILES as sentences representing chemical structures. It is trained in three stages: (i) self-supervised pretraining, (ii) domain-specific pretraining, and (iii) fine-tuning for molecular property prediction (Fig. 1a). Initially, a transformer-based language model learns chemical language from a large compound corpus (pretraining). Then, it undergoes further self-supervised training on domain-specific compounds, optionally using data augmentation. Finally, the model is fine-tuned through supervised training for specific tasks. Throughout, SMILES representations are processed into chemical "words" as input for ChemLM (Fig. 1b).

(i) *Language-model pretraining*: Pretraining, a key step in transfer learning, involves training the model on millions of samples before fine-tuning for a specific task. Masked language modeling (MLM) randomly masks input tokens, training the model to predict them using the surrounding context. ChemLM was first trained on 10 million ZINC compounds using MLM, following BERT[25]. Unlabeled tokenized SMILES data were used to learn compound representations, creating the ChemLM base model that encodes the syntax and semantics of chemical compounds.

(ii) *Domain adaptation for the language modeling*: In this stage, the pretrained model is further trained on domain-specific, unlabeled data, refining its ability to capture task-specific structures and improving performance[28,29]. Domain adaptation addresses domain shift differences in data distributions between pretraining and target tasks, which can hinder generalization. This is crucial since ChemLM, trained on millions of diverse compounds, must perform well on structurally similar molecules. In natural language processing (NLP), domain adaptation resembles fine-tuning, achieved through continued pretraining or smaller task-specific datasets, as shown by Gururangan et al.[30]. Similarly, MLM-based domain adaptation has been effective in NLP[31], demonstrating that unsupervised task-specific training enhances transformer models. To counter limited domain-specific data, we applied SMILES enumeration[32], generating additional representations by reordering atoms (Supplementary Algorithm 1). This is a

computationally efficient augmentation method to expand the whole dataset by several factors. Since the model is trained unsupervised using MLM, no information leaks into the evaluation phase.

(iii) *Supervised fine-tuning of the transformer language model network*: In the final phase, the trained model undergoes supervised fine-tuning. To prevent overfitting, we deploy early stopping in addition to techniques in model development, e.g., L2 regularization. Instead of freezing the transformer's layers and fine-tuning only the classification head, we choose to unfreeze all of them and further fine-tune them to optimize performance. The attention maps, spread across various layers of a transformer model trained on chemical compounds, can be utilized to demonstrate how different chemical tokens interact in creating the final language model-based embedding of these compounds (Supplementary Fig. 1).

### Architecture optimization

Hyperparameters significantly impact deep learning models, thus, we analyzed key parameters in ChemLM and transformers for molecular property prediction. Using Optuna, we optimized the augmentation number, hidden layers, attention heads, and embedding types (Supplementary Table 1) and assessed their influence via Optuna's f-ANOVA test (Fig. 2). A crucial factor was the augmentation number in domain adaptation training, representing alternative SMILES forms. We tested randomized SMILES between 0 and 100 and found that high values (80-100) were consistently selected, leading to a linear increase in training time (Supplementary Table 2). Following BERT[25], we explored optimal embeddings by combining layer weights through summation or averaging, either in the last layer or across multiple layers. We also compared using the first token versus all tokens, as the first token encapsulates sequence information and receives the most attention[25,33]. Embeddings type strongly influenced performance (Fig. 2), whereas attention heads and layer count had minimal impact. The final optimized hyperparameters for each task are detailed in Supplementary Table 3.

### ChemLM identifies potent pathoblockers for *P. aeruginosa*

In drug discovery, oftentimes, a very limited number of compounds are available, substantially fewer than those included on commonly used benchmark datasets for chemical property prediction tasks. To assess the value of ChemLM model for a real-world drug discovery problem, we employed it to identify potent pathoblockers compounds acting against *P. aeruginosa* (Fig. 3a), which is one of the priority pathogens identified by the World Health Organization, often characterized by multidrug resistance[6]. The class of compounds that we focused on disrupts the quorum-sensing (QS) machinery of *P. aeruginosa*[34–38] (PqsR Inverse Agonists 2018 Ref. No. WO2020007938A1 (EP18181475), New PqsR Inverse Agonist 2020 (EP20150104), and Novel PqsR Inverse Agonists 2020 Ref. No. WO2021136805A1 (EP20150119)) using a compound library of 219 structures with varying potency. The drug target is the QS receptor and transcription factor PqsR[39].

Small molecular compounds acting on PqsR via an inverse agonistic mode-of-action reduce the production of several virulence factors such as the toxin pyocyanin. The initial hit already impaired pyocyanin production with a potency in the double-digit micromolar range and was characterized by a trifluoromethyl-pyridine fragment[37]. A lead generation campaign via structure-guided fragment growing was initiated, which yielded five QS inhibitor classes with substantially increased potency[34–36] (Fig. 3a) and retaining this fragment motif. The inhibitor classes are described in more detail in peer-reviewed journals or patent applications (Supplementary Table 4). We use the $IC_{50}$ to measure drug potency, which is the inhibitor concentration needed to inhibit a biological process in vitro by 50%. Highly potent compounds have an $IC_{50}$ of <500 nM. For the five classes, the number of compounds and their potencies vary considerably; from 2 to 107, and include between 0 and 71 highly potent compounds.

To rigorously evaluate the performance of the ChemLM model, we devised a challenging scenario. Given the substantial variation in the

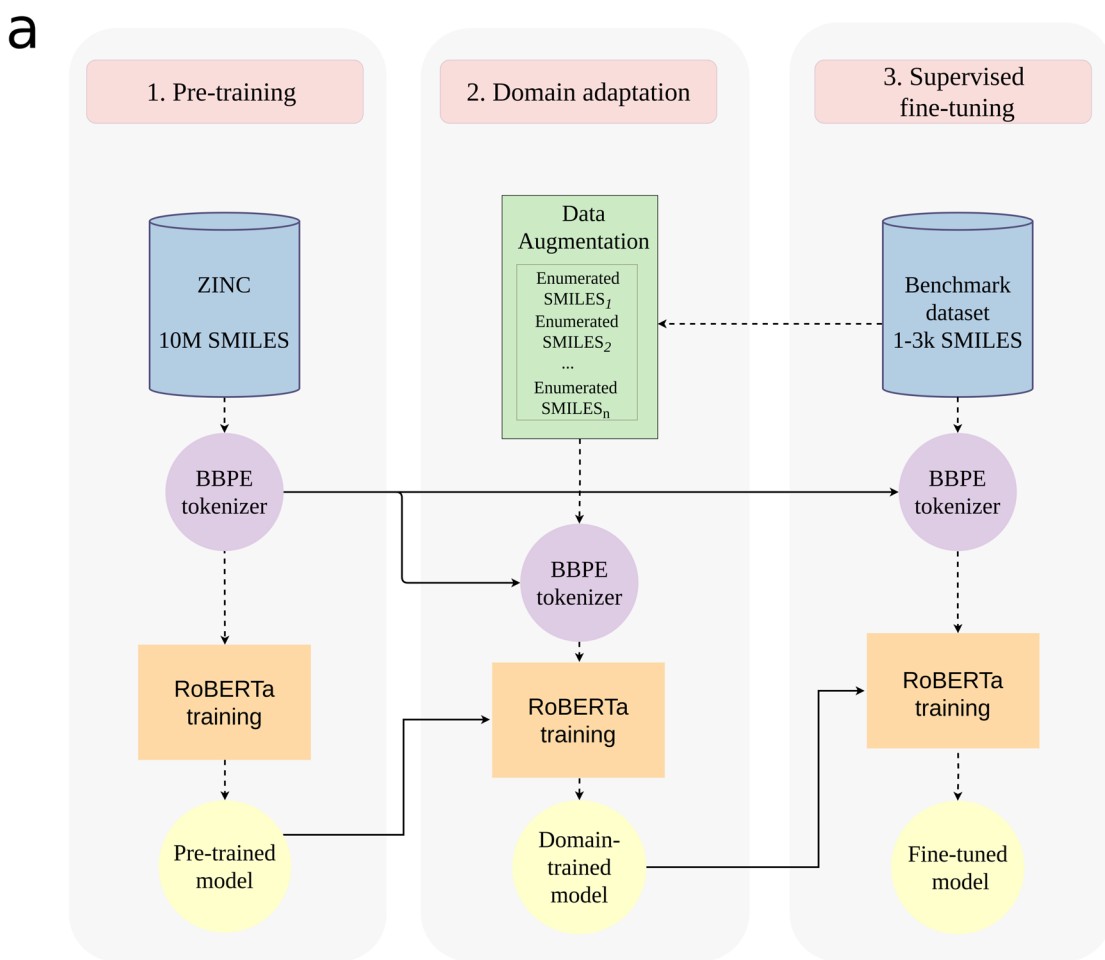

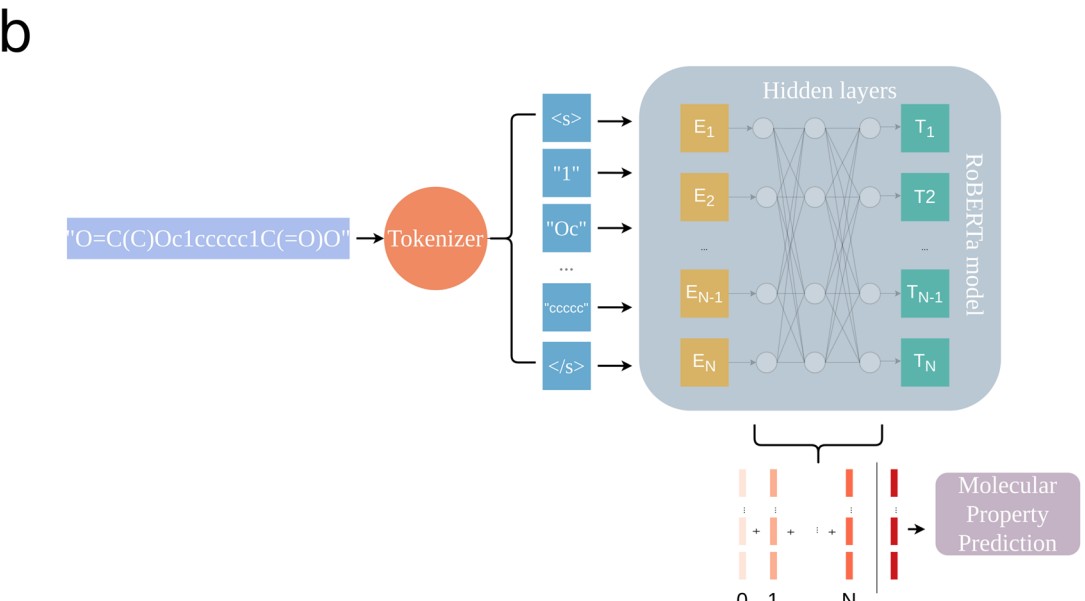

**Fig. 1 | The ChemLM training strategy. a** Training stages of the ChemLM model. All the trained models are represented by circular shapes, BBPE models are in purple and RoBERTa is in yellow. Procedures like training, augmentation, and prediction are indicated with rectangles. The dashed line indicates the flow of information within a training stage, whereas the solid line describes the transfer of knowledge from one training stage to another. **b** An example that indicates how a SMILES string is processed and treated by the ChemLM transformer model. Firstly, it gets tokenized and special tokens are added to the sequence. Then, these are fed into the model and at the end, the sum of weights from the hidden layers is used to make predictions.

**Fig. 2 | Importance of hyperparameters in model's performance during hyperparameter optimization using the validation data of each dataset.** The examined hyperparameters are: the embeddings type, the number of attention heads and hidden layers, and the augmentation number.

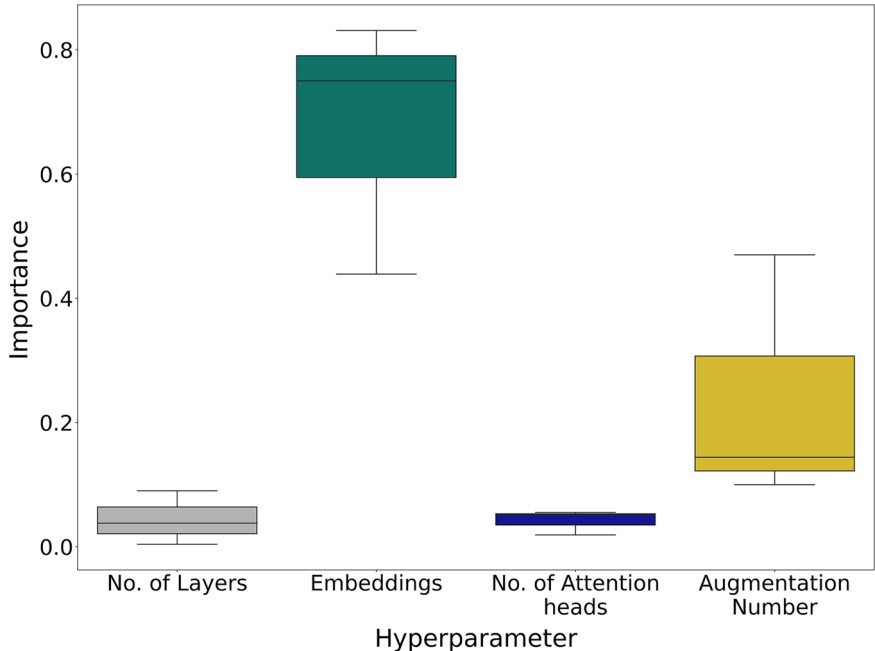

number of compounds per class in the compound library, we pursued an alternative approach to partition the data into more similarly sized folds. We employed ward linkage hierarchical clustering on the ChemLM embeddings and partitioned the library into five sets of chemically similar compounds, resulting in a more even distribution (Supplementary Table 5). Specifically, we organized the compound library by grouping compounds into these folds based on ChemLM's embeddings similarity. This approach ensures that compounds with chemical similarity, even if they belong to different structural classes, are kept together within the same fold, as opposed to using the initial structural classes. This strategy helps prevent information leakage during model training and introduces a demanding challenge for the ChemLM model. Subsequently, we conducted the third stage of model training using the SMILES representations of compounds from four of the folds. The compounds from the remaining fold were then classified as highly potent or not. This process was repeated for each set of folds (Fig. 3b) and the same hyperparameters were used for all models (Supplementary Table 3).

For model assessment, we compared ChemLM to leading graph neural networks and language models. Graph models included MPNN[40], GCNN[41], and GAT[42], using DeepChem (v2.6.0) with default settings. Language models included MolFormer[43], MolBERT[44], and ChemBERTa[23] using the "PubChem10M_SMILES_BPE_180k" model from Hugging Face, pre-trained on 10 million SMILES from PubChem. Furthermore, approaches that did not provide pretrained models or codes were excluded. ChemLM's training extends methods used in ChemBERTa and vanilla ChemLM by adding a second training stage on task-specific data with augmented SMILES representations and hyperparameter optimization.

In comparison to these models, ChemLM achieved the highest median of macro-averaged F1-scores (0.90), which is almost 30% more than that of the second-best model (MPNN; Fig. 3b, Supplementary Table 6). The same applies to all the evaluation metrics we examined. Moreover, its performance in identifying highly potent pathoblockers is quite high, as the F1-score for that class in each of the five folds consistently ranges from above 0.83 to a maximum of 0.92 in all folds (Supplementary Table 7). Most notably, ChemLM demonstrates consistency when compared to other models, which either fail or perform poorly on this task in certain folds. In addition, ChemBERTa achieved a median F1 score of 0.33 across folds (Supplementary Table 6 and Fig. 3b), faced challenges particularly for the positive class, recording an F1-score of only 0.17 for that class in the 4th fold (Supplementary Table 7). These results highlight the value of the optimized

ChemLM for identifying highly potent compounds for an application with a very limited number of compounds available for a task-specific training scenario.

## Optimizing ChemLM substantially improves performance

We evaluated ChemLM's performance on binary classification tasks for molecular property prediction, comparing it to the same models as earlier across three benchmark datasets (Supplementary Table 8). Datasets were split using DeepChem's splitter[45] to maintain class distribution across training (70%), validation (10%), and test (20%) sets. Training parameters for graph neural networks, including epochs and learning rate, were optimized via grid search within the DeepChem framework.

First, a ChemLM vanilla model was trained without using a domain adaptation phase or hyperparameter optimization. In its architecture, 12 layers and attention heads were included and pooling as the type of embeddings (Supplementary Table 3). A second model, ChemLM domain-adapted, was then trained on domain-specific data, with augmented SMILES representations and no hyperparameter optimization using the same architecture as ChemLM vanilla. Finally, for the ChemLM domain-adapted and optimized model, all the hyperparameters were optimized and in addition, we unfroze the model's layers for fine-tuning in the task-specific training.

To assess performance, we primarily report the macro-averaged F1-score, as a balanced reflection of the performance of the models across both positive and negative classes, irrespective of their sizes. For further details on the model performances, we also provide the accuracy, AUC, precision, and recall values for the individual classes. In terms of the macro-average F1-score metric, the optimized ChemLM was among the top performers in benchmark evaluation (Fig. 4). It performed substantially better than the graph-based models, with an improvement of up to 0.25 in macro-averaged F1-score on the ClinTox dataset relative to the second-best performing model (0.91 vs. 0.66; Table 1). ChemBERTa had an F1-macro averaged score of 0.9 and 0.87 on the ClinTox and BBBP datasets, respectively (Table 1 and Supplementary Table 9), performing slightly less well than ChemLM on these (0.92 and 0.88, respectively). On the BACE dataset, ChemBERTa's performance was notably weaker, achieving a macro-averaged F1 score of 0.69, compared to 0.8 for ChemLM (Supplementary Table 10). Compared to MolBERT, which is also based on transformers, we observed a very similar performance for two of the datasets, 0.88 versus 0.89

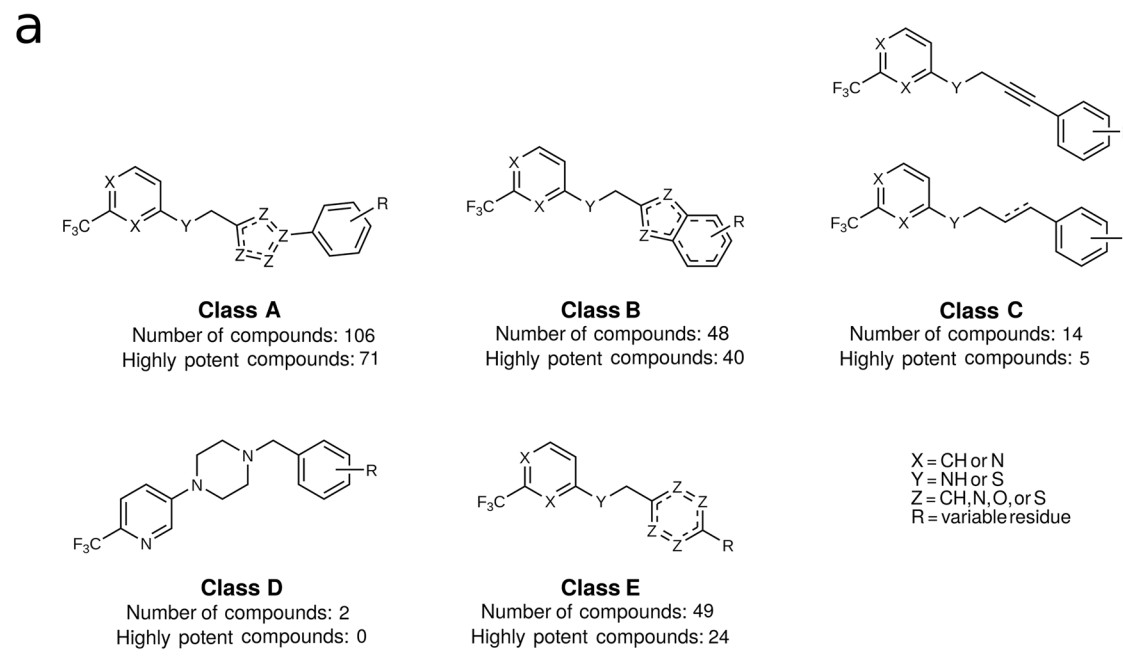

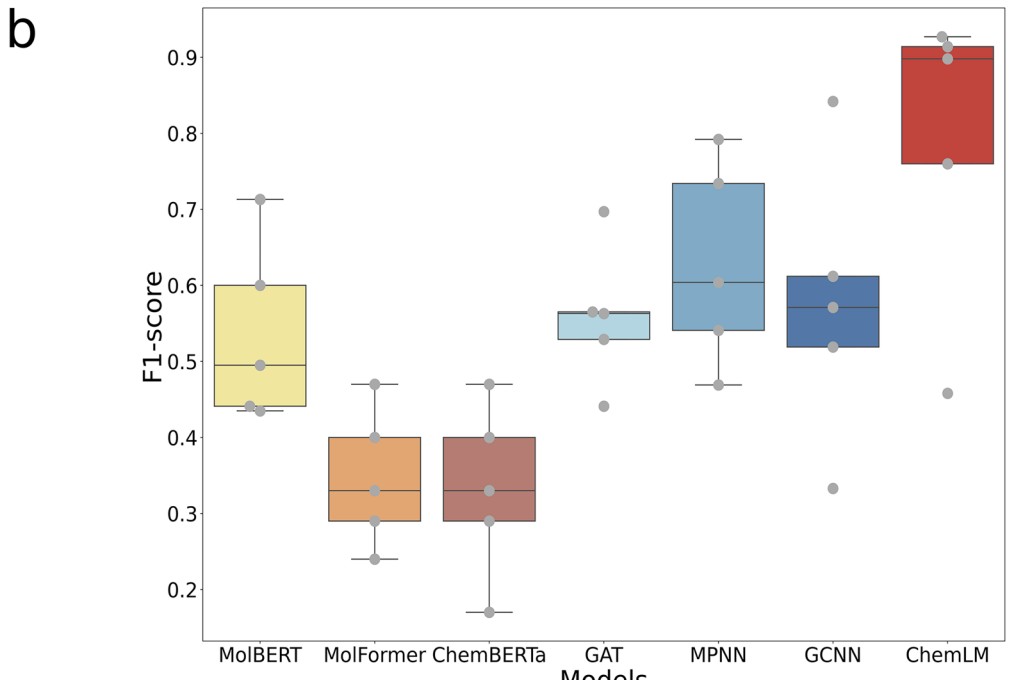

**Fig. 3 | Description of experimental data. a** Chemical structure and the number of compounds per class. **b** Performance comparison of ChemLM with graph neural networks and transformer-based approaches in 5-fold validation for experimental compounds on *Pseudomonas aeruginosa*. The graph neural networks (blue) are graph attention transformers (GAT)[42], message-passing neural networks (MPNN)[40], and graph convolutional neural networks (GCNN)[41]. MolBERT[44], MolFormer[43], and ChemBERTa[23] are transformer-based approaches. The ChemLM model is noted in red. Gray dots represent the F1-scores achieved by the model across the five folds.

on BBBP and 0.80 versus 0.81 on BACE, respectively (Supplementary Tables 9 and 10). On the contrary, MolFormer achieved a slightly higher F1-score on the BBBP dataset, 0.92 versus 0.8 (Supplementary Table 9), through performing worse with other evaluation metrics and demonstrated a notably better performance on the BACE dataset, 0.8 versus 0.9 (Supplementary Table 10). However, ChemLM substantially outperformed them on the ClinTox dataset with the macro-averaged F1-score increasing by 36% from MolBERT's 0.67 to 0.91 (Table 1). This performance improvement of the optimized ChemLM on the ClinTox dataset is primarily due to the

substantially lower performance of all other models on the positive class (Supplementary Table 11), ranging from 0.22 to 0.38 versus 0.84 for the ChemLM model. Notably, the positive class of the ClinTox data set has the lowest number of samples of all data sets and the largest degree of class imbalance. Similarly to what we observed for the experimental pathoblocker dataset, all other models tended to not identify the few positive samples correctly for this dataset (Supplementary Tables 6 and 11).

We also observed a substantial improvement between the vanilla and the domain-adapted ChemLM models, demonstrating the benefits of

**Fig. 4 | Performance of ChemLM and state-of-the-art models with the macro averaged F1-score on the test data sets of the benchmark data.** ChemLM and its variations are compared with state-of-the-art models. Red diamonds represent the mean macro-averaged F1-score for each model across the three datasets.

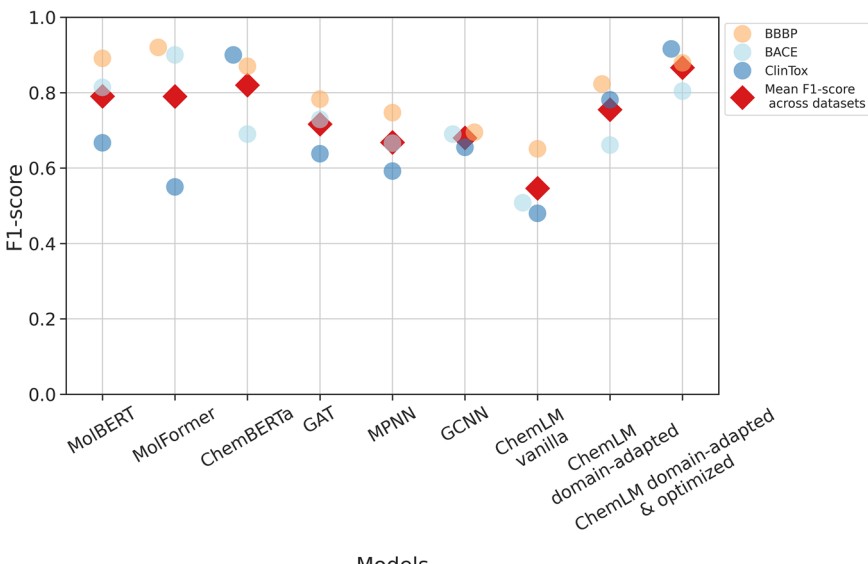

### Table 1 | Comparison of ChemLM on ClinTox dataset with its simpler versions, and state-of-the-art models in more evaluation metrics

| Model | F1 | AUC | Precision | Recall | Accuracy |
|---|---|---|---|---|---|
| MolFormer | 0.55 | 0.7 | 0.89 | 0.7 | 0.95 |
| MolBERT | 0.67 | 0.67 | 0.66 | 0.67 | 0.92 |
| ChemBERTa | 0.9 | 0.84 | 0.99 | 0.84 | 0.98 |
| MPNN | 0.64 | 0.59 | 0.87 | 0.59 | 0.94 |
| GAT | 0.59 | 0.57 | 0.72 | 0.57 | 0.93 |
| GCNN | 0.66 | 0.61 | 0.78 | 0.61 | 0.94 |
| ChemLM vanilla | 0.48 | 0.50 | 0.46 | 0.50 | 0.93 |
| ChemLM domain-adapted | 0.82 | 0.75 | 0.98 | 0.75 | 0.96 |
| ChemLM domain-adapted & optimized | 0.92 | 0.86 | 0.99 | 0.86 | 0.98 |

The macro-averaged score is reported for each metric.

adding the domain adaptation stage and the data augmentation within it. The performance improvements range from 15% for the BACE dataset, from 0.51 to 0.80 for the macro-averaged F1-score, to up to 30% for the ClinTox dataset, from 0.48 to 0.92 with this metric. The complete evaluation of the models for these datasets can be found in the Supplementary material (Supplementary Tables 9 and 10).

#### ChemLM embeddings reflect molecular properties of chemical compounds

To evaluate whether ChemLM's embeddings reflect molecular properties relevant to drug efficacy, we analyzed their relationship to chemical properties. Since transformer models are self-supervised, no inherent correlation is expected before fine-tuning. To this end, we calculated the median ratio of distances between property values and embedding vectors for randomly selected compounds from the BBBP dataset, comparing it to a shuffled property dataset. This analysis covered three key physicochemical properties: molecular weight, quantitative estimate of drug-likeness (QED), and polar surface area. Using 200 randomly selected compounds, we generated embeddings over 100 rounds to produce a ratio distribution (see the "Methods" section, Fig. 5a). Results showed significantly lower ratios in ChemLM's space compared to the shuffled labels (one-sided *t*-test, Table 2, Supplementary Table 12), indicating that ChemLM embeddings effectively capture meaningful molecular properties.

To explore this behavior in similar models, we analyzed the local relationships of the embeddings of the MolBERT and ChemLM to these molecular properties, by calculating the ratio using the embedding and property distances of the drug pairs from the randomly subsampled molecules of the BBBP dataset (Fig. 5b, Table 3). Violin plots of the property differences show that both model embeddings have a similar relationship to the properties, with comparable median and median absolute deviation values. These findings confirm that both ChemLM's and MolBERT's embeddings capture and reflect the molecular properties.

To qualitatively assess our results, we visualized the embeddings of molecules in a two-dimensional space using UMAP. This approach allowed us to determine whether compounds are encoded in meaningful embeddings in the ChemLM model, aligning chemicals with similar physicochemical properties in close proximity, while maintaining the global structure of the data distribution. We applied this technique for the previously assessed molecular properties including the aromatic rings in our evaluation (Fig. 5c). For all properties, we observed a gradual change of these properties in this space, indicating that molecules with similar properties tend to possess similar embedding values.

#### Discussion

In this study, we introduce ChemLM, a language model for molecular property prediction, incorporating key innovations in chemical language modeling. The first is an additional training stage, where the model undergoes self-supervised training on domain-specific compound representations extending the standard pretraining-finetuning approach[23,24]. Similar unsupervised domain and task adaptation strategies have proven effective in NLP across deep learning architectures[29,46] benefiting chemical transformer models. This adaptation phase enhances the model's understanding of chemical associations, improving predictive performance, especially for tasks with limited domain-specific data. The second innovation is data augmentation on domain datasets, generating alternative sequence representations to increase training instances, a particularly beneficial technique for chemical tasks with small, imbalanced datasets. In this work, we demonstrate the effectiveness of this approach, combined with a domain adaptation stage, which leads to substantial performance improvements, especially for classification tasks involving small, imbalanced datasets.

We comprehensively assessed the ChemLM model on suitable benchmark datasets for molecular property prediction; the BACE (inhibition of the BACE-1 enzyme), the BBBP (blood–brain barrier penetration), and the ClinTox dataset (clinical toxicity) originating from MoleculeNet.

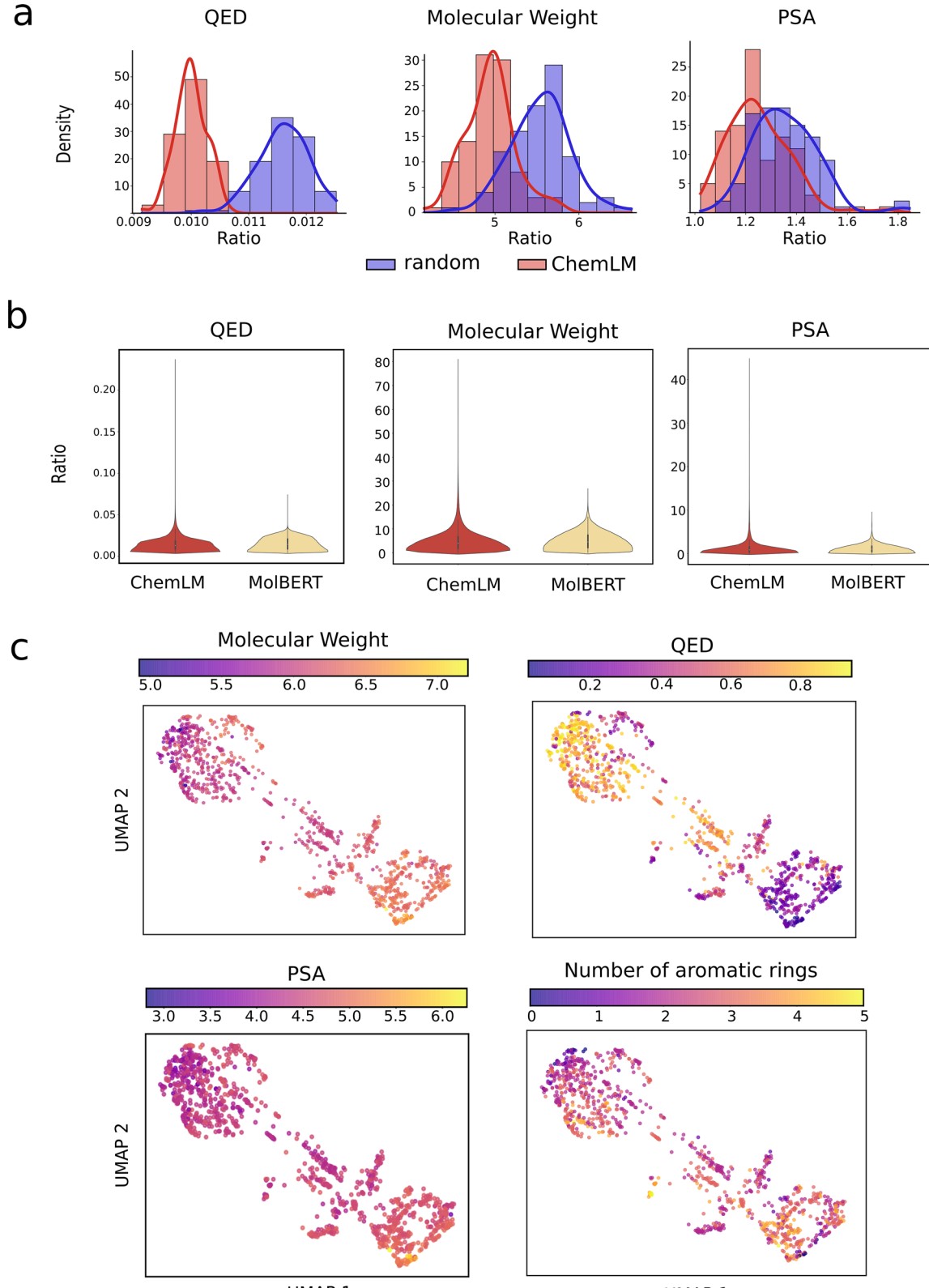

**Fig. 5 | Intrinsic evaluation of ChemLM. a** Distribution plot showing the ratio of property differences relative to embedding distances for ChemLM and random space across three molecular properties. The properties are a quantitative estimation of drug-likeness (QED), molecular weight, and polar surface area (PSA). **b** Violin plots showing the ratio between the ChemLM and MolBERT models for randomly sampled molecule pairs from the BBBP dataset. **c** UMAP plots of molecular properties. They demonstrate the distribution of molecular properties. Each dot represents a molecule in the dataset. The molecular weight and PSA values have been scaled using the natural logarithm, while the actual values have been used for QED and the number of aromatic rings.

Across all datasets, ChemLM demonstrated a substantial performance gain of up to 20% relative to graph neural networks. These results suggest that a transformer-based approach can surpass the performance of leading Graph Neural Network architectures. When compared to other language processing models, such as MolBERT and MolFormer, ChemLM showed substantial performance improvements, particularly on the highly imbalanced ClinTox dataset, where it outperformed the other methods by more than 20% in F1-score. These findings highlight the effectiveness of ChemLM in accurately identifying the positive class within imbalanced datasets.

As a real-world test case, we also evaluated ChemLM on its ability to identify compounds targeting *Pseudomonas aeruginosa*, a hospital-acquired pathogen known for its multi-drug resistance. The model demonstrated substantial performance gains in identifying potent pathoblocker compounds effective against *Pseudomonas aeruginosa* from a chemical compound library, specifically targeting the transcription factor PqsR. To thoroughly evaluate the model's ability to make predictions for structurally diverse candidate molecules, we stratified the dataset of experimental compounds based on structural similarities for cross-validation. In this evaluation, ChemLM achieved a 30% performance improvement over the second-best model for this task. The model also achieved a high F1-score for the positive class (highly potent pathoblockers) across all folds, demonstrating its ability to generalize effectively and maintain high consistency in performance. These results suggest that the performance gains offered by ChemLM could significantly aid in identifying relevant drug compounds for pharmacological applications. Further development of ChemLM could expand its capabilities from predicting active compounds to estimating activity levels and suggesting potential compound structures via generative models, thereby supporting a wide range of future experimental data analyses.

Compared to standard training, ChemLM showed significant performance gains with expanded self-supervised training across all benchmarks. Hyperparameter optimization identified embeddings as the most influential factor. During domain adaptation, incorporating multiple molecular representations further improved performance. These findings provide valuable insights for future hyperparameter tuning.

To assess the relationship between ChemLM embeddings and drug-relevant properties, we computed the ratio of property differences to embedding distances. Median ratio values were significantly lower than those in a randomly shuffled space. We also analyzed property distributions within ChemLM and MolBERT embeddings, where smaller ratios indicated more coherent mappings. Overall, the evaluation showed that transformer models encode molecular properties meaningfully, providing robust representations across multiple properties.

In summary, we describe ChemLM, an optimized chemical language encoder model designed to predict the molecular properties of chemical compounds. Our evaluation across multiple datasets demonstrated substantial performance improvements achieved through self-supervised training on domain-specific data and data augmentation, resulting in enhanced accuracy in molecular property prediction and the creation of a chemically meaningful encoding space. Additionally, hyperparameter optimization boosted the model's performance. Together, these findings highlight the potential of transformer models in advancing chemical research. Notably, the model's primary achievement lies in its successful application to real-world data and predictive challenges. It excelled at identifying potent pathoblockers against *P. aeruginosa* using a very limited amount of training data, underscoring its promise to accelerate drug discovery efforts in the future.

## Methods

### Data description

We used two types of datasets to train and evaluate the model's performance. The first is the ZINC (v15) database, a public collection of millions of chemical compounds[47]. We retrieved the SMILES representations of the molecules and used them in the pretraining stage of the ChemLM model. The second ones were three benchmark datasets from MoleculeNet[48] for predicting the physicochemical properties of molecules (Supplementary Table 8). BACE's target class indicates binding results for a set of inhibitors to $\beta$-secretase 1. The blood–brain barrier penetration dataset (BBBP) is a collection of compounds from a study about compounds' brain barrier permeability in which class labels indicate penetration or non-penetration. ClinTox includes compounds that can be used for the tasks of FDA approval status and clinical trial toxicity. Here, we evaluated the models on the second task.

### Tokenization using byte pair encoding (BPE)

One critical step is tokenizing SMILES, treating each string as a sentence of tokens. ChemLM uses byte-level byte pair encoding (BBPE)[49] for this, as recommended for RoBERTa. Initially a compression method, BPE[50] assigns new symbols to frequent character pairs, enabling a hybrid word/character-level tokenization. Another advantage is the user-defined vocabulary size, determining token count and influencing tokenization. Since SMILES has a smaller vocabulary than natural languages, we explored different sizes trained on ZINC—from 10,000, as Wang et al. suggested[49], to 2058, including special tokens, and used this setting in further experiments. To learn byte sequences, a BBPE tokenizer was trained on ZINC, a large SMILES corpus making it adaptable for various applications and datasets.

**Transformers.** ChemLM is based on transformers, a deep learning architecture originally designed with an encoder–decoder structure[22]. At the core of multi-head attention lies the concept of self-attention, which focuses on generating improved representations of the sequence elements (tokens) by considering their interactions with neighboring elements. This self-attention mechanism is utilized within multi-head attention to

### Table 2 | Median ratio values and their *p*-value for each molecular property

| Molecular property | ChemLM | Random space | Ratio | *p*-value |
|---|---|---|---|---|
| Molecular weight | 4.951 | 5.56 | 0.89 | 7.48e−32 |
| QED | 0.01 | 0.012 | 0.83 | 5.05e−60 |
| Polar surface area | 1.24 | 1.35 | 0.93 | 8.74e−14 |

Low median values were observed for the ratio of property distances to embedding distances of most properties and a relatively stable ratio of the ChemLM to permuted data constant. *P*-values were calculated using a one-tailed *t*-test to assess whether the means of the ChemLM ratio distribution was significantly higher than the one generated by random permutation of the molecular property labels.

### Table 3 | Statistical measures of property differences for ChemLM and MolBERT

| Statistical measure | QED | | Polar surface area | | Molecular Weight | |
|---|---|---|---|---|---|---|
| | ChemLM | MolBERT | ChemLM | MolBERT | ChemLM | MolBERT |
| Max value | 0.23 | 0.07 | 44.59 | 9.45 | 80.34 | 26.38 |
| Median value | 0.008 | 0.01 | 0.89 | 1.02 | 3.94 | 4.51 |
| Standard deviation | 0.008 | 0.007 | 1.63 | 0.88 | 4.58 | 3.48 |
| MAD | 0.005 | 0.006 | 0.52 | 0.58 | 2.28 | 2.51 |

enable the model to attend to multiple views of the sequence interactions simultaneously, resulting in more expressive and informative representations. Thus, each layer of the encoder includes a multi-headed attention sublayer and a position-wise fully connected feed-forward network followed by normalization layers. In attention in transformer models, each token of the sequence is associated with two real-valued vector representations: a key vector, $k$ from the input embedding space and a value vector, $v$ from the output embedding space. These vectors can be either randomly initialized or pretrained. The query vector $q$ represents the sequence element for which one wants to obtain a new representation and must belong to the same space as the key vectors. To calculate a new representation for the entire sequence, the key, $k$, query, $q$, and value $v$ vectors are calculated using dot multiplication of the embedding with the corresponding learned weight matrices. Matrix multiplications are deployed to leverage efficiency and parallelization. Embeddings, query, key, and value vectors are packed into matrices, $X$, $K$, $Q$, and $V$. Attention is calculated as described in equation 1, in which $d_k$ stands for the dimension of vector $k$.

$$\text{Attention}(Q, K, V) = softmax\left(\frac{QK^{\mathrm{T}}}{\sqrt{d_k}}\right)V \qquad (1)$$

Instead of using a single attention mechanism, researchers introduced multihead attention. Its benefit to the model lies in the information that is captured from different representation subspaces at different positions.

$$\text{MultiHead}(Q, K, V) = Concat(\text{head}_1, \ldots, \text{head}_h)W^0, \qquad (2)$$

where each $\text{head}_i$ is equal to

$$\text{head}_i = Attention(QW_i^Q, KW_i^K, VW_i^V) \qquad (3)$$

and $W_i$ is the weight matrix.

Matrices $W_i^Q \in R^{d_{\text{model}} \times d_k}$, $W_i^K \in R^{d_{\text{model}} \times d_k}$, $W_i^V \in R^{d_{\text{model}} \times d_V}$ and $W^O \in R^{hd_V \times d_{\text{model}}}$ where $h$ are the parallel attention layers are learned weights. Another key element is the addition of Position Encoding to provide information about the position of tokens in the sequence and counteract the absence of recurrent or convolutional elements.

$$\text{PE}_{(pos,2i)} = \sin\left(\frac{pos}{10,000^{\frac{2i}{d_{\text{model}}}}}\right) \qquad (4)$$

$$\text{PE}_{(pos,2i+1)} = \cos\left(\frac{pos}{10,000^{\frac{2i}{d_{\text{model}}}}}\right) \qquad (5)$$

In formulas 4 and 5, pos stand for the position in the sequence, $d_{\text{model}}$ for the dimension of the output embedding space and $i$, for the embedding index. This architecture offers several advantages, addressing sequence model limitations. The self-attention mechanism captures long-term dependencies by modeling interactions between distant tokens. Transformers are highly scalable, handling variable-length inputs independently at each position. Their parallelizable design enables efficient computation, reducing training and inference time. Additionally, transfer learning allows pretraining on unlabeled data before fine-tuning for specific tasks.

RoBERTa was chosen from autoencoder models. Based on BERT, it uses learnable position embeddings instead of sinusoidal encodings (see formulas 4 and 5). With an average SMILES sequence length of 45 tokens, RoBERTa is well-suited. Language models have various training tasks, like next-sentence prediction. RoBERTa masks tokens and learns to predict them from context, a process called masked language modeling. This approach helps the model grasp SMILES syntax and grammar, making RoBERTa ideal for this application.

## ChemLM implementation and training

To implement ChemLM, HuggingFace[51] (version 0.0.8) was used to configure and train the RoBERTa model for the first training stages. A combination of Huggingface and PyTorch[52] (version 1.6) was used for the supervised fine-tuning. In addition, scikit[53] (0.24.1) was deployed for hierarchical clustering and evaluation metrics and RDKit[54] (v2020.09.1.0) to produce the molecular properties for the intrinsic evaluation.

The ChemLM model was trained using MLM in the first two training stages, pretraining and domain adaptation. The domain adaptation training stage used multiple SMILES representations for each molecule. These representations were generated using SMILES enumeration, a data augmentation technique for SMILES strings[32]. We experimented with different numbers of augmentations (Supplementary Table 2) per molecule to find the best-performing one during the hyperparameter optimization. All training was performed using an NVIDIA t4 GPU.

## Intrinsic evaluation

**Quantitative evaluation.** To assess the degree to which the learned compound embeddings of the ChemLM model reflect properties such as the molecular weight, polar surface area, and the quantitative estimate of drug-likeness that are relevant for drug efficacy, we assessed the continuity of their representations in the embeddings. Thus, we used the ratio ($r$) of property difference to embedding distance. ChemLM was compared to the properties of a data set with randomly shuffling assigned molecular properties. Specifically, we calculated the properties for a subset of compounds from the BBBP dataset using RDKit and examined the ratio as described as (Eq. (6))

$$r = \frac{d_f(f(e_1), f(e_2))}{d_e(e_1, e_2)} \qquad (6)$$

where $f$ is the property value, $d_f$ the absolute difference of these values for the embeddings $e_1$, $e_2$, $r$ is the ratio and $d_e$ the Euclidean distance of the embeddings.

The ratio is used to understand how property values change with molecular embedding variations. A low, stable ratio indicates predictable changes. Embeddings were derived from the final layer's weights. To compare ChemLM with a shuffled random space, we subsampled 200 compounds for 100 rounds from 1200 BBBP compounds. A one-tailed $t$-test (Scipy v1.8.0) assessed statistical significance, testing whether ChemLM's ratio is lower than random, and computed the $p$-value using the 'greater' alternative. For MolBERT comparison, we calculated the ratio for all molecule pairs in the selected BBBP compounds for both models.

**Qualitative evaluation.** In addition to the quantitative evaluation of the trained space, we projected the 768-dimensional vectors of molecule embeddings to 2D space using the UMAP algorithm. The molecular properties examined are molecular weight, polar surface area (PSA), quantitative estimate of drug-likeness (QED), and the number of aromatic rings. To make the distribution of the properties more evident, we scaled the values of the first two properties using the natural logarithm. In this way, we can visually inspect the distribution of the aforementioned properties.

## Data availability

Data is available at https://github.com/hzi-bifo/ChemLM/data to reproduce benchmark experiments and intrinsic evaluation. The generated data used for analysis and to produce figures is located at https://github.com/hzi-bifo/ChemLM/results. Experimental data have been presented in peer-reviewed journals or patent applications and are detailed in Supplementary Table 4.

## Code availability

Code is available at https://github.com/hzi-bifo/ChemLM. The code for the experimental part is not included, as the dataset is internal. Models are available at https://huggingface.co/gkallergis.

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

## Acknowledgements

G.K. gratefully acknowledges financial support from the Lower Saxony Ministry for Science and Culture for the doctoral program "Drug Discovery and Cheminformatics for New Anti-Infectives (iCA)".

## Author contributions
G.K., E.A., and A.C.M. conceived the study. G.K. implemented the software. E.A. supervised the code development. A.C.M. and E.A. supervised the work. G.K., E.A., and A.C.M. have written the article. A.H. and M.E. have shared the experimental dataset, advised and guided the work on the corresponding part. M.E. has contributed to writing, too. F.K. advised on the intrinsic evaluation and provided feedback. All authors have reviewed the article.

## Funding

## Competing interests
The authors declare no competing interests.
