## [Transparent Peer Review file · Communications Chemistry]

Domain adaptable language modeling of chemical compounds identifies potent pathoblockers for *Pseudomonas aeruginosa*

Corresponding Author: Professor Alice McHardy

Version 0:

Reviewer comments:

Reviewer #1

(Remarks to the Author)

Overall this manuscript introduces a new attribute prediction model. the addition of a domain adaptation training phase, which subtly improves the final prediction performance, but there are still the following question for the conclusion.

1. would the same prediction performance be achieved under the same conditions without the addition of a domain-adaptive training phase? how to prove?
2. what is the composition and source of the data for the second stage?
3. has multi-task learning been considered for multiple attribute prediction?
4. for predicting potent pathoblockers for *Pseudomonas aeruginosa*, are the fine-tuned compounds private? what is the confidence level of the data?

Reviewer #2

(Remarks to the Author)

This paper proposes an accurate language model-based molecular properties prediction model, namely "ChemLM." Here are some questions for the authors:

1. What differentiates ChemLM from previous language model-based approaches such as ChemBERTa (2020), which utilized the RoBERTa model and tokenized SMILES strings? ChemBERTa demonstrated considerable performance in predicting molecular properties. <https://arxiv.org/abs/2010.09885>
2. Could you please clarify the definition of domain adaptation? In the manuscript, domain adaptation seems to overlap with fine-tuning. Traditionally, domain adaptation involves removing source domain differences from multiple datasets, including acquisition parameters, scanners, cameras, and ethnic characteristics. However, fine-tuning utilizes the pre-trained model's parameters for target-specific (downstream) tasks. Please clarify this point.
3. How was data augmentation conducted? The manuscript lacks details and logic regarding data augmentation. It would be helpful to include explanations using pseudocode or diagrams.
4. Additionally, what is the difference between SMILES enumeration? Please explain the difference between SMILES enumeration and your data augmentation methods.
5. In Figure 1A, SMILES tokens are tokenized at the character level. However, in the manuscript, SMILES are tokenized using 10k tokens of output from the BBPE tokenizer. This could confuse readers' understanding. Please modify the figure accordingly.
6. Figure 2 does not have section A. However, Figure 2A was presented on page 7. Please verify this inconsistency.
7. Most details are described in supplementary tables, severely degrading reader understanding. If the performances were presented in the main text, readability could be improved.
8. It is difficult to understand the Lipschitz constant comparison between ChemLM and random output. Isn't it natural for ChemLM to have a low Lipschitz constant? Additionally, for pair comparison, ChemLM should be compared to other models such as MolBERT.
9. For baseline models, MolBERT, GAT, MPNN, and GCNN were utilized. There are many language model-based molecular property prediction models; therefore, demonstrating ChemLM's performance could be more persuasive.
 - a. Wang, S., Guo, Y., Wang, Y., Sun, H., & Huang, J. (2019, September). Smiles-bert: large scale unsupervised pre-training

for molecular property prediction. In Proceedings of the 10th ACM international conference on bioinformatics, computational biology and health informatics (pp. 429-436).

b. Honda, S., Shi, S., & Ueda, H. R. (2019). Smiles transformer: Pre-trained molecular fingerprint for low data drug discovery. arXiv preprint arXiv:1911.04738.

c. Maziarka, Ł., Danel, T., Mucha, S., Rataj, K., Tabor, J., & Jastrzębski, S. (2020). Molecule attention transformer. arXiv preprint arXiv:2002.08264.

d. Fabian, B., Edlich, T., Gaspar, H., Segler, M., Meyers, J., Fiscato, M., and Ahmed, M. Molecular representation learning with language models and domain-relevant auxiliary tasks. arXiv preprint arXiv:2011.13230, 2020

10. Please explain the one-tailed t-test utilized for Table 1.

11. The token size set to 10k followed the original paper. However, the original paper's implementation was targeted towards natural language, while SMILES has a significantly different environment. This statement needs further explanation.

12. In the Methods section, Transformer subsection, readability is severely degraded. For example, "(i) a key vector (k)" contains many brackets. It could be replaced with mathematical representations or clarified using other methods.

Version 1:

Reviewer comments:

Reviewer #1

(Remarks to the Author)

I accept the revised manuscript, I have no further comments

Reviewer #2

(Remarks to the Author)

The authors have almost resolved my previous concerns, and I really appreciate it. However, the remaining issue regarding the differences between ChemLM and ChemBERTa is still not clearly explained. Here are my questions:

Major Comments

1. It is a bit confusing whether ChemLM refers to (1) a training strategy of pretraining-domain adaptation, continual pretraining, and fine-tuning-or (2) a BERT-like encoder that can understand molecular sequences. If the focus is on (1), it would be beneficial to demonstrate the robustness of this strategy with other encoders, such as ChemBERTa and MolBERT. On the other hand, if the main focus is on (2), I think the authors should include prediction performances using ChemBERTa for comparison, as it seems very similar to ChemLM. I suggest that for ChemBERTa, three versions of comparisons would be favorable: vanilla ChemBERTa, domain-adapted ChemBERTa, and domain-adapted & optimized ChemBERTa.

Minor Comments

1. Please check the abbreviations, such as "masked language modeling." It should be "MLM," as the authors defined it as MLM when it first appeared.

Nature Communications Chemistry

Dear reviewers,

we would like to express our sincere gratitude to you for their thoughtful and constructive feedback to our manuscript. We appreciate the time and effort invested in providing insightful comments that have allowed us to substantially improve the manuscript.

We have carefully reviewed each of the reviewers' comments and we are confident that we have systematically addressed all reviewers' comments, providing explanations, clarifications, and revisions as required. We are dedicated to ensuring that our manuscript meets high standards of quality and rigor, and we welcome the opportunity to further refine our work in light of the reviewers' feedback.

Briefly, the revised manuscript includes the following changes:

- **Reviewer 1:** The reviewer inquired about the added value of the domain-adaptation training stage and the composition of the training data. In response, we further clarified how that stage improves the performance of ChemLM and which data have been used for its training.
- The reviewer suggested describing the availability of the experimental data. In response, we outlined the publications and the patents in which the structure of each chemical class has been described.
- **Reviewer 2:** The reviewer suggested including additional language processing approaches in our comparisons and proposed some models that we could use. To address these comments, we included one more language processing approach, called Molformer on benchmark scenarios and experimental compounds, and further clarified our choice of models in the manuscript.

- The reviewer suggested comparing ChemLM to MolBERT in qualitative analysis. We implemented this suggestion by including more statistical measures and figures to further outline how the models map the compounds on chemical space.
- The reviewer suggested clarifying details of the tokenization method in ChemLM in Fig 1b, which we addressed, as suggested.
- The reviewer also suggested making changes to our text to improve readability. In response, we moved tables and figures (Table 1, Figure 5) and enhanced the suggested parts of the manuscript.

In conclusion, with these changes and the additional analysis that we included comprehensively addressed the reviewers' concerns and substantially improved our manuscript. We believe that our work makes an important contribution to the field of molecular property prediction, showing that the language-model realized with ChemLM is effective with benchmark scenarios and in particular for small, imbalanced data sets generated by medicinal chemists, such as an experimental pathoblocker data set against *Pseudomonas aeruginosa*. As small sample sizes of a few dozen candidate compounds are common in experimental drug discovery, this innovation holds substantial value for the field of computational chemistry, with the potential to facilitate drug development.

Please do not hesitate to contact us if you have any other questions.

Sincerely,

Prof. Dr. Alice McHardy
Helmholtz Centre for Infection Research

Reviewer #1 (Remarks to the Author):

Overall this manuscript introduces a new attribute prediction model. the addition of a domain adaptation training phase, which subtly improves the final prediction performance, but there are still the following question for the conclusion.

1. would the same prediction performance be achieved under the same conditions without the addition of a domain-adaptive training phase? how to prove?

We thank the reviewer for the thoughtful comment, indeed this is correct. We describe in the manuscript two language models, "ChemLM vanilla" and "ChemLM domain-adapted". The first model has only two training stages (pretraining and fine-tuning), whereas the second model, "ChemLM domain-adapted", has three training stages, including the domain-adaptation phase as the only difference to ChemLM vanilla. We specify this (p. 11, line 315) as follows, "*First, a ChemLM vanilla model was trained without using a domain adaptation phase or hyperparameter optimization. In its architecture, 12 layers, and attention heads were included, and pooling as the type of embeddings (Supplementary Table A3). A second model, ChemLM domain-adapted, was then trained on domain-specific data, with augmented SMILES representations, and no hyperparameter optimization took place, using the same architecture as ChemLM vanilla*".

Across benchmark data sets, the domain-adaptation phase indeed substantially, between 13 - 34%, improves the predictive performance in comparison to the same model trained on the same data without this phase across benchmark data sets (p. 11, Fig 4). We describe the implications in the discussion (p. 17 line 452), "..., we noticed substantial improvement in ChemLM's performance due to our methodological improvements across all benchmark datasets, such as the addition of domain adaptation stage." and we have reported the results on p. 12 in Table 1 and on Appendix as well, on page 26 Supplementary Tables A9 and A10.

2. what is the composition and source of the data for the second stage

Thank you for pointing out this need for clarification of the text. The data for the second training stage come from the specific task dataset that is also utilized in the third training stage. To further clarify this, we extended the description in the result section (p. 5 subsection "(ii) Domain adaptation for the language modeling"): "*In this stage, the pre-trained model is further trained on the domain-specific data used in the third training stage, without using the labels*". The difference in the data that are used in these training stages lies in the composition, only the SMILES sequences are used in the second stage, whereas SMILES and the prediction labels are used in the third training stage. This is stated in the results section, p. 4 line 144 is stated: "*Then, the model is further trained in a self-supervised manner on domain-specific compounds. Optionally, the training instances are extended with a data augmentation algorithm to cover multiple views on the chemicals. In*

the last step the model is fine-tuned by supervised training on the domain-specific compounds for a given task.”

3. has multi-task learning been considered for multiple attribute prediction?

We appreciate the reviewer's valuable suggestion regarding the deployment of multi-task learning. In our work, we used ChemLM for predicting individual attributes for different benchmarking tasks, therefore multiple attribute prediction has not been required. However, indeed, this is a relevant problem in drug discovery processes and we are currently working on multi-task learning for multiple attribute predictions on further data sets from experimental collaborators.

4. for predicting potent pathoblockers for *Pseudomonas aeruginosa*, are the fine-tuned compounds private? what is the confidence level of the data?

We thank the reviewer for this important comment. For clarification, we have added a table to the manuscript (p. 32, Supplementary Table A4), which specifies that a large portion of the compounds are described in peer-reviewed journals or patent applications.

Table A4 Table of publications and patents detailing each structural class in the experimental data.

Structural class	Publication	Patent
A	Schütz et al [45], Abdelsamie et al [49]	WO2020007938A1 WO2021136805A1 WO2021136803A1
B		WO2020007938A1 WO2021136803A1
C	Schütz et al [46], Zender et al [48]	
D		
E	Hamed et al [47], Zender et al [48]	WO2020007938A1 WO2021136805A1 WO2021136803A1

Reviewer #2 (Remarks to the Author):

This paper proposes an accurate language model-based molecular properties prediction model, namely "ChemLM." Here are some questions for the authors:

1. What differentiates ChemLM from previous language model-based approaches such as ChemBERTa (2020), which utilized the RoBERTa model and tokenized SMILES strings? ChemBERTa demonstrated considerable performance in predicting molecular properties. <https://arxiv.org/abs/2010.09885>

We thank the reviewer for the valuable comment. To clarify the differences of our method to ChemBERTa, we added the following text to the results (p. 9 line 280): *"... Of the available language processing models, we compared ChemLM to MolFormer and MolBERT, which represent the currently leading models with demonstrated performance gains over other models, such as ChemBERTa Methodologically, the ChemLM model extends the approach used by other models such as ChemBERTa and the "vanilla" ChemLM by incorporating a second training stage on task-specific data, by using an augmented dataset of SMILES representations in this stage, and hyperparameter optimizations..."*

2. Could you please clarify the definition of domain adaptation? In the manuscript, domain adaptation seems to overlap with fine-tuning. Traditionally, domain adaptation involves removing source domain differences from multiple datasets, including acquisition parameters, scanners, cameras, and ethnic characteristics. However, fine-tuning utilizes the pre-trained model's parameters for target-specific (downstream) tasks. Please clarify this point.

We appreciate the reviewer's comment and have clarified this with the following additions to the manuscript (p. 5 line 169): *"Domain adaptation addresses the problem of differing data distributions between the source and target datasets, known as domain shift, which can lead to poor generalization. This issue is relevant here because our pretrained model is trained on a diverse database of millions of compounds, while it is applied to tasks involving molecules with similar chemical structures. Therefore, domain adaptation is necessary to ensure the model performs well and generalizes effectively. In natural language processing (NLP), domain adaptation is similar to fine-tuning. For instance, in the context of transformers, domain adaptation can be achieved through continued pre-training or using smaller task-specific data, as supported by Gururangan et al. Additionally, masked language modeling has been used for domain adaptation in NLP in a similar way to ChemLM, as noted by Arefyev et al."* Thus, we believe ChemLM aligns with the principles and objectives of domain adaptation in the NLP community.

Gururangan, Suchin, Ana Marasović, Swabha Swayamdipta, Kyle Lo, Iz Beltagy, Doug Downey, and Noah A. Smith. 2020. "Don't Stop Pretraining: Adapt Language Models to Domains and Tasks." arXiv [cs.CL]. arXiv. <http://arxiv.org/abs/2004.10964>.

Arefyev, Nikolay, Dmitrii Kharchev, and Artem Shelmanov. 2021. "NB-MLM: Efficient Domain Adaptation of Masked Language Models for Sentiment Analysis." In Proceedings of the 2021 Conference on Empirical Methods in Natural Language Processing, edited by Marie-Francine Moens, Xuanjing Huang, Lucia Specia, and Scott Wen-Tau Yih, 9114–24. Online and Punta Cana, Dominican Republic: Association for Computational Linguistics.

3. How was data augmentation conducted? The manuscript lacks details and logic regarding data augmentation. It would be helpful to include explanations using pseudocode or diagrams.

We thank the reviewer for this helpful suggestion. As suggested, we added the pseudocode for the data augmentation to the manuscript (p. 31, Supplementary Algorithm 1):

Algorithm 1 Pseudocode for SMILES enumeration algorithm

```
1: nm ← []
2: number_augmentations ← 100
3: counter ← 0
4: for all smiles ∈ smiles_list do
5:   for counter ← 0 to number_augmentations – 1 do
6:     atom_numbers ← smiles.get_number_atoms()
7:     m ← shuffle(atom_numbers)
8:     nm.extend(renumberAtoms(m, atom_numbers))
9:   end for
10: end for
```

4. Additionally, what is the difference between SMILES enumeration? Please explain the difference between SMILES enumeration and your data augmentation methods.

We apologize for the lack of clarity; this refers to the same technique. We have further clarified the statement in p. 5, line 180. It now reads:

"To avoid having too little domain-specific data for model training, we performed 'SMILES enumeration.' This process involves creating a specified number of additional compounds for each individual SMILES representation by reordering the atoms, following the approach described by Bjerrum et al. (see Supplementary Algorithm 1)."

Bjerrum, Esben Jannik. 2017. "SMILES Enumeration as Data Augmentation for Neural Network Modeling of Molecules." *arXiv [cs.LG]*. arXiv. <https://doi.org/10.1021/ci034143r>.

5. In Figure 1A, SMILES tokens are tokenized at the character level. However, in the

manuscript, SMILES are tokenized using 10k tokens of output from the BBPE tokenizer. This could confuse readers' understanding. Please modify the figure accordingly.

We appreciate the reviewer's attention to detail. To clarify, tokens are created to the Byte-level BPE (BBPE) tokenizer, which creates the tokens used in the following without further tokenization steps. We assume that his impression arose as the previous version of Fig. 1B only displayed the first tokens created, which were mostly individual characters, however, this is not the case for the complete list. To clarify we have updated Fig. 1B to the following:

6. Figure 2 does not have section A. However, Figure 2A was presented on page 7. Please verify this inconsistency.

We thank the reviewer for the careful assessment. We removed the "Section A" from the reference to figure 2.

7. Most details are described in supplementary tables, severely degrading reader understanding. If the performances were presented in the main text, readability could be improved.

Thank you for the thoughtful suggestion. In response, we have moved the former Supplementary Table A10 with performance metrics from the Supplement to the main text, and include it as Table 1 in page 12 in the main manuscript. Moreover, we have moved the former Supplementary Fig. 2 as Fig. 5a in page 15 of the main text. Additionally, we have revised the performance evaluation sections, clarifying our use of metrics, providing the numbers for the major performance metric, the macro-averaged F1-score, in the main text in all places.

8. It is difficult to understand the Lipschitz constant comparison between ChemLM and random output. Isn't it natural for ChemLM to have a low Lipschitz constant? Additionally, for pair comparison, ChemLM should be compared to other models such as MolBERT.

To clarify, we added the following to the respective section (p. 13, line 366): *“Transformer models are trained in a self-supervised fashion and before finetuning them for a supervised classification task, there is no natural correlation with further properties necessarily expected.”*

As suggested further we also performed an analysis comparing ChemLM to MolBERT in terms of meaningfulness of the embedding space for different molecular methods, see below.

We have furthermore revised the corresponding section to improve clarity as follows:

Results section (p 13, line 363):

“To assess whether the compound embeddings generated by ChemLM are reflective of the underlying molecular properties relevant for drug efficacy, we assessed how the models embeddings relate to the chemical properties of the compounds. To this end, we calculated the median of the ratio of the distances between property values and embeddings vectors for randomly selected compounds from the BBBP dataset and compared it to the values calculated for the same data set with randomly shuffled molecular property values.

We applied this analysis to three relevant physicochemical properties: molecular weight, quantitative estimate of drug-likeness (QED), and polar surface area. We used the chemical properties of 200 randomly selected chemical compounds and their embeddings generated by ChemLM for 100 rounds to produce a ratio distribution (Methods, Fig. 5a). The results of our analysis demonstrated that, for all properties, ChemLM space’s ratios exhibited significantly lower values compared to the randomly shuffled label space (one-sided t-test, Table 2). This consistent behavior suggests that ChemLM effectively maps molecules in an informative and meaningful manner.

Table 2 Median ratio values and its p-value for each molecular property.

Molecular Property	ChemLM	Random Space	Ratio	p -value
Molecular Weight	4.951	5.56	0.89	7.5e-32
QED	0.01	0.012	0.83	5.05e-60
Polar surface area	1.24	1.35	0.93	8.74e-14

Low median values were observed for the ratio of property distances to embeddings distances of most properties, and a relatively stable ratio of the *ChemLM* to permuted data constant. P-values were calculated using a one-tailed t-test to assess whether the means of the *ChemLM* ratio distribution was significantly higher than the one generated by random permutation of the molecular property labels.

To explore this behavior in similar models, we analyzed the local relationships of the embeddings of the MolBERT and ChemLM models to these molecular properties, by calculating the ratio using the embedding and property distances of the drug pairs from the randomly subsampled molecules of the BBBP dataset (Fig. 5b, Table 3). Violin plots of the property differences show that both model embeddings have a similar relationship to the properties, with comparable median and median absolute deviation values. These findings confirm that both ChemLM's and MolBERTs embeddings capture and reflect the molecular properties.

Table 3 Statistical measures of property differences for ChemLM and MolBERT.

Statistical measure	QED		Polar surface area		Molecular Weight	
	ChemLM	MolBERT	ChemLM	MolBERT	ChemLM	MolBERT
Max value	0.23	0.07	44.59	9.45	80.34	26.38
Median value	0.008	0.01	0.89	1.02	3.94	4.51
Standard deviation	0.008	0.007	1.63	0.88	4.58	3.48
MAD	0.005	0.006	0.52	0.58	2.28	2.51

Fig. 5 Intrinsic evaluation of ChemLM. a) Distribution plots showing the ratio of property differences relative to embedding distances for *ChemLM* and random space across three molecular properties. The properties are quantitative estimation of drug-likeness (QED), molecular weight and polar surface area (PSA). b) Violin plots showing the ratio between the *ChemLM* and MolBERT models for randomly sampled molecule pairs from the BBBP dataset. c) UMAP plots of molecular properties. They demonstrate the distribution of molecular properties. Each dot represents a molecule in the dataset. The molecular weight and PSA values have been scaled using the natural logarithm, while the actual values have been used for QED and the number of aromatic rings.

Discussion (p. 17 line 464)

"To quantify the relationship between the embeddings generated by ChemLM and chemical properties relevant for drug efficacy, we calculated the ratio of the property differences to the distances between their embedding vectors. Our analysis showed that, for all properties, the median ratio values were significantly lower compared to a space with randomly shuffled labels. Additionally, we assessed for ChemLM and MolBERT the distribution of these molecular properties in the embedding space, with smaller values indicating a more meaningful mapping with respect to that properties. Overall, the intrinsic evaluation indicated that transformer models such as ChemLM and MolBERT provide a chemically meaningful encoding of the space for multiple molecular properties."

Methods (p 21, line 597)

"..To assess the degree to which the learned compound embeddings of the ChemLM model reflect properties such as the molecular weight, polar surface area, and the quantitative estimate of drug likeness that are relevant for drug efficacy, we assessed the continuity of their representations in the embeddings. Thus, we used the ratio (r) of property difference to embeddings distance. ChemLM was compared to the properties of a data set with randomly shuffling assigned molecular properties. Specifically, we calculated the properties for a subset of compounds from the BBBP dataset using RDKit and examined the ratio as described in equation 6,

$$r = \frac{d_f(e_1, e_2)}{d_e(e_1, e_2)}, \quad (6)$$

where f is the property value, d_f the absolute difference of these values for the embeddings e_1, e_2 , r is the ratio, and d_e the Euclidean distance of the embeddings. The rationale behind utilizing the ratio lies in our intention to understand how the values of properties change with variations in the embeddings of molecules. A low and stable ratio signifies that the properties change predictably, and consistently for different input compounds. We utilized embeddings generated from the weights of the final layer. To compare ChemLM with the random space that is generated by shuffling the property values, we randomly subsampled 200 chemical compounds for 100 rounds from 1200 randomly selected compounds of the BBBP dataset.

To assess whether these results are statistically important, we also perform a one-tailed t-test on the distributions of ChemLM 's, and random space's ratio. A one-tailed t-test was used to calculate the p-value to assess whether the null hypothesis, that the k of our space is lower than the random one's, is true. Scipy (v. 1.8.0) was used to perform the t-test, and calculate the p-value using as an alternative argument, the 'greatest'. For comparison with the MolBERT model, we used all the molecule pairs of the randomly selected compounds of BBBP dataset for both MolBERT and ChemLM to calculate the ratio for each model. "

9. For baseline models, MolBERT, GAT, MPNN, and GCNN were utilized. There are many language model-based molecular property prediction models; therefore, demonstrating ChemLM's performance could be more persuasive.

a. Wang, S., Guo, Y., Wang, Y., Sun, H., & Huang, J. (2019, September). Smiles-bert: large scale unsupervised pre-training for molecular property prediction. In Proceedings of the 10th ACM international conference on bioinformatics, computational biology and health informatics (pp. 429-436).

b. Honda, S., Shi, S., & Ueda, H. R. (2019). Smiles transformer: Pre-trained molecular fingerprint for low data drug discovery. arXiv preprint arXiv:1911.04738.

c. Maziarka, Ł., Danel, T., Mucha, S., Rataj, K., Tabor, J., & Jastrzębski, S. (2020). Molecule attention transformer. arXiv preprint arXiv:2002.08264.

d. Fabian, B., Edlich, T., Gaspar, H., Segler, M., Meyers, J., Fiscato, M., and Ahmed, M. Molecular representation learning with language models and domain-relevant auxiliary tasks. arXiv preprint arXiv:2011.13230, 2020

We thank the reviewer for highlighting these related preprints. To clarify, we included the following to the results section (p. 9, line 280):

"... Of the available language processing models, we compared ChemLM to MolFormer and MolBERT, which represented the currently leading models with demonstrated performance gains over other models, such as ChemBERTa. Furthermore, if the trained model or necessary code was unavailable for benchmarking, such as for Smiles-bert, we were unable to include it in our assessment. ..."

Specifically, in relation to the above mentioned methods a-d:

- a) does not provide the pretrained files for the models, which would require us to recreate their entire model on our own, which take substantial time and resources (GPUs) to produce. Furthermore, this model was published in 2019 and has since been superseded by many more recent methods, including those that we benchmark against.
- b) lacks the vocabulary file that is needed for tokenization.
- c) provides a pretrained model, but the preprint is also four years old without any publication that we could identify, thus it has been superseded by many more recent methods. It also lacks the evaluation and training code which is partially replaced by pseudocode.
- d) The method described in (d) is MolBERT, which is included already in our publication.

Furthermore, we also evaluated another approach called MolFormer (Ross *et al*, 2022) and was published in Nature Machine Intelligence, being one of the most sophisticated models in the field. It is a language approach that uses RoFormer, a transformer model that relies on rotational embeddings. We report its performance in the following tables (Table 1 and

Supplementary Tables A6, A7, A9, A10, A11) and figures (Figures 3b, 4). In addition, we modified the text (p. 12 line 336) as follows:

"...MolFormer achieved a slightly higher F1-score on the BBBP dataset, though performing worse with other evaluation metrics, and demonstrated a notably better performance on the BACE dataset. However, ChemLM substantially outperformed both methods on the imbalanced ClinTox dataset, with the macro-averaged F1-score increasing by 36% from MolBERT's 0.67 to 0.91 (Table 1). ..."

(p. 17 line 458)" *... Additionally, it performed similarly to MolBERT and MolFormer, both language processing approaches, with the exception of one dataset (BACE); however, ChemLM substantially surpassed them in performance on the ClinTox dataset, by 25% and 36% in F1-score respectively. ..."*

Ross, Jerret, Brian Belgodere, Vijil Chenthamarakshan, Inkit Padhi, Youssef Mroueh, and Payel Das. 2022. "Large-Scale Chemical Language Representations Capture Molecular Structure and Properties." *Nature Machine Intelligence* 4 (12): 1256–64.

We have modified the following tables and figures.

- **Experimental data (p. 33)**

Table A6 Performance comparison of property prediction models over the test set of a 5-fold cross-validation setting over the experimental dataset. The macro-averaged score is reported for each metric.

Model	F1	AUC	Precision	Recall	Accuracy
MolFormer	0.45	0.49	0.47	0.47	0.73
MolBERT	0.495	0.5	0.553	0.6	0.714
MPNN	0.604	0.592	0.661	0.591	0.789
GAT	0.563	0.583	0.635	0.583	0.714
GCNN	0.571	0.567	0.575	0.568	0.651
ChemLM	0.899	0.900	0.900	0.900	0.900

The median metric value of each model is demonstrated.

Table A7 Predictive performance of *ChemLM* and state-of-the-art models on the positive class (highly potent pathoblockers).

Hierarchical Folds	ChemLM	MPNN	GAT	GCNN	MolBERT	MolFormer
1	0.92	0.94	0.92	0.91	0.87	0.45
2	0.83	0.8	0.82	0.77	0.78	0.45
3	0.90	0.72	0.62	0.63	0.70	0.73
4	0.89	0.33	0.00	0.75	0.00	0.42
5	0.89	0.75	0.73	0.00	0.60	0.67

The F1-score is reported as evaluation metric in this this table.

a

b

Fig. 3 Description of experimental data: (a) Chemical structures, and number of compounds per class. (b) Performance comparison of *ChemLM* with graph neural networks and transformers - based approaches in 5-fold validation for experimental compounds on *Pseudomonas aeruginosa*.

Benchmark data

Fig. 4 Performance of ChemLM and state-of-the-art models with the macro averaged F1-score on the test data sets of the benchmark data. ChemLM and its variations are compared with state-of-the-art models. The graph neural networks (blue) are GAT (Graph Attention Transformers)[16], MPNN (Message Passing Neural Networks)[14], and GCNN (Graph Convolutional Neural Networks)[54]. MolBERT [60] and MolFormer [55] (in yellow) are transformer-based approaches. ChemLM models are noted in red. ChemLM demonstrates equal or better performance to the state-of-the-art models.

ClinTox(p.12)

Table 1 Comparison of *ChemLM* on ClinTox dataset with its simpler versions, and state-of-the-art models in more evaluation metrics. The macro-averaged score is reported for each metric.

Model	F1	AUC	Precision	Recall	Accuracy
MolFormer	0.55	0.7	0.89	0.07	0.95
MolBERT	0.667	0.671	0.662	0.671	0.915
MPNN	0.638	0.593	0.870	0.593	0.938
GAT	0.592	0.566	0.719	0.566	0.928
GCNN	0.655	0.614	0.784	0.614	0.935
ChemLM vanilla	0.480	0.500	0.462	0.500	0.925
ChemLM domain-adapted	0.823	0.750	0.980	0.750	0.962
ChemLM domain-adapted & optimized	0.916	0.864	0.989	0.864	0.979

BBBP (p. 34)

Table A9 Comparison of *ChemLM* on BBBP dataset with its simpler versions, and state-of-the-art models in more evaluation metrics. The macro-averaged score is reported for each metric.

Model	F1	AUC	Precision	Recall	Accuracy
MolFormer	0.92	0.71	0.89	0.71	0.86
MolBERT	0.891	0.888	0.895	0.888	0.928
MPNN	0.783	0.788	0.778	0.788	0.841
GAT	0.747	0.711	0.847	0.711	0.85
GCNN	0.695	0.664	0.820	0.664	0.828
ChemLM vanilla	0.689	0.674	0.72	0.674	0.799
ChemLM domain-adapted	0.823	0.811	0.837	0.81	0.87
ChemLM domain-adapted & optimized	0.879	0.885	0.874	0.884	0.912

BACE (p 34)

Table A10 Comparison of *ChemLM* on BACE dataset with its simpler versions, and state-of-the-art models in more evaluation metrics. The macro-averaged score is reported for each metric.

Model	F1	AUC	Precision	Recall	Accuracy
MolFormer	0.9	0.91	0.91	0.91	0.91
MolBERT	0.814	0.814	0.813	0.814	0.816
MPNN	0.729	0.733	0.731	0.733	0.729
GAT	0.666	0.704	0.769	0.704	0.680
GCNN	0.69	0.692	0.731	0.692	0.71
ChemLM vanilla	0.508	0.554	0.603	0.553	0.584
ChemLM domain-adapted	0.661	0.662	0.674	0.662	0.673
ChemLM domain-adapted & optimized	0.804	0.803	0.804	0.803	0.805

Positive class prediction of benchmark datasets (p 34)

Table A11 Comparison of *ChemLM* on benchmark datasets with state-of-the-art models in prediction of the positive class using F1-score as evaluation metric.

Model	ClinTox	BACE	BBBP
MolFormer	0.55	0.90	0.92
MolBERT	0.38	0.8	0.96
MPNN	0.31	0.72	0.90
GAT	0.22	0.73	0.91
GCNN	0.35	0.61	0.90
ChemLM domain-adapted & optimized	0.84	0.79	0.94

10. Please explain the one-tailed t-test utilized for Table 1.

We addressed this, as suggested. The legend for what was previously Table 1 (now Table 2 on page 13) has been expanded as follows: "Low median values were observed for the ratio of property distances to embeddings distances of most properties, and a relatively stable ratio of the ChemLM to permuted data constant. P-values were calculated using a one-tailed t-test to assess whether the means of the ChemLM ratio distribution was significantly higher than the one generated by random permutation of the molecular property labels."

11. The token size set to 10k followed the original paper. However, the original paper's implementation was targeted towards natural language, while SMILES has a significantly different environment. This statement needs further explanation.

The reviewer is absolutely correct, thank you for pointing this out. In fact, we experimented with different token sizes, ranking from 10k to 2,058, and the final setting was set to 2,058 tokens, as the initial vocabulary of the SMILES language is more limited compared to natural language, which we had not specified in the text, so the comment was very valuable.

To clarify, we adapted the text in p. 18, line 518 to "As the *initial vocabulary of the SMILES language is more limited than natural languages, we explored different settings of the vocabulary sizes, ranging from 10,000 as suggested by Wang et al, to 2,058 training on the ZINC database, and using the 2,058 token setting, including special tokens, for further experiments.*

12. In the Methods section, Transformer subsection, readability is severely degraded. For example, "(i) a key vector (k)" contains many brackets. It could be replaced with mathematical representations or clarified using other methods.

Thank you for this suggestion. We have improved the text to the following (p. 19 line 536):

"In attention in transformer models, each token of the sequence is associated with two real-valued vector representations: a key vector, k from the input embedding space, and a value vector, v from the output embedding space. These vectors can be either randomly initialized or pre-trained. The query vector q represents the sequence element for which one wants to obtain a new representation, and must belong to the same space as the key vectors. To calculate a new representation for the entire sequence, k , q , and v vectors are calculated using dot products of the embedding with the corresponding learned weight matrices."

#Reviewer 2

Major Comments

1a. It is a bit confusing whether ChemLM refers to (1) a training strategy of pretraining-domain adaptation, continual pretraining, and fine-tuning-or (2) a BERT-like encoder that can understand molecular sequences.

Thank you for the constructive comment. We have carefully gone over the manuscript, to ensure these points are well explained. We outline below where we have added further clarifications and where relevant content is listed. We outline that it is (2) a new encoder benefitting from (1) a specific training strategy in model generation and performing well across a range of benchmarking scenarios in all sections of the manuscript:

Abstract:

p. 1 line 31: *"Here, we introduce ChemLM, a transformer language model-based approach for this task"*

p. 1 line 32: *"ChemLM leverages self-supervised domain adaptation on chemical molecules to enhance its predictive performance."*

Introduction:

p. 4 line 113: *"Here, we describe ChemLM, a language modeling-based approach for efficient transfer learning for chemical compounds. "*

p. 4 line 119: *"With this, we aimed for an approach **model** that can be applied for real-world datasets of experimental compounds that comprise limited training samples/compounds."*

p. 4 line 121: *"We assessed whether language models' training using domain adaptation, which allows us to adapt the pre-trained model on further data from the target domain, enhances the model's predictive ability."*

Results:

p. 4 line 133: *"ChemLM is a transformer-based method **model** that processes molecules' SMILES as sentences representing the chemical structures. "*

p. 4 line 134: *"ChemLM was trained with a three stage approach (Fig. 1a), consisting of (i) a self-supervised pre-training stage, (ii) a secondary domain-specific pre-training and then, (iii) a fine-tuning stage for the supervised classification in molecular property prediction tasks."*

p. 7 line 209: "As a key part of the ~~ChemLM~~ **training**, we also investigated the optimal augmentation number for domain adaptation training"

p. 7 line 233: "To assess the value of the ChemLM **model** for a real-world drug discovery problem, ..."

p. 7 Fig. 1 caption: "The ChemLM ~~approach~~ **training strategy**. a) Training stages of the ChemLM model. ... b) An example that indicates how a SMILES string is processed and treated by the ChemLM transformer model. ..."

p. 7 line 256: "To rigorously evaluate the performance of ChemLM **model**, we devised a challenging scenario."

p. 9 line 286: "Methodologically, the ChemLM ~~model~~ training approach extends the ~~approach~~ ones used for training of other models such as the 'vanilla' ChemLM and ChemBERTa by incorporating a second training stage on task-specific data, using an augmented dataset of SMILES representations in this stage and hyperparameter optimizations."

p. 12 line 351: "This performance improvement of the optimized ChemLM on the ClinTox dataset is primarily due to the substantially lower performance of all other models on the positive class (Supplementary Table A11), ranging from 0.22 to 0.38 versus 0.84 for the ChemLM ~~approach~~ **model**."

Discussion:

p. 16 line 408: "In this study, we describe ChemLM, a language modeling-based ~~approach~~ for efficient transfer learning in the field of molecular property prediction for chemical compounds."

p. 16 line 415: "Similar to the training scheme applied for ChemLM, which includes an additional, unsupervised training stage on the task specific chemical structure data, including unsupervised domain and task adaptation training schemes have proven their robustness in natural language processing across multiple deep learning architectures and prediction tasks [59, 60]. Taken together, these findings indicate benefits of this approach also for improving chemical compound transformer models."

p. 16 line 433: "We comprehensively assessed the ChemLM **model** on suitable benchmark datasets for molecular property prediction; ..."

p. 17 line 475: "To quantify the relationship between the embeddings generated by the ChemLM **model** ... "

p. 17 line 487: "In summary, we describe ChemLM, an optimized **chemical language encoder model** for predicting the molecular properties of chemical compounds."

p. 17 line 495: "Notably, the model's primary achievement lies in its successful application to real-world data and predictive challenges."

1b. If the focus is on (1), it would be beneficial to demonstrate the robustness of this strategy with other encoders, such as ChemBERTa and MolBERT. On the other hand, if the main focus is on (2), I think the authors should include prediction performances using ChemBERTa for comparison, as it seems very similar to ChemLM. I suggest that for ChemBERTa, three versions of comparisons would be favorable: vanilla ChemBERTa, domain-adapted ChemBERTa, and domain-adapted & optimized ChemBERTa.

We appreciate the thoughtful suggestion. To clarify that the employed training strategy will likely be beneficial for other transformer models, including those trained on chemical compound structures such as ChemBERTa and MolBERT, we have added the following (page 16 line 415):

"Similar to the training scheme applied for ChemLM, which includes an additional, unsupervised training stage on the task specific chemical structure data, including unsupervised domain and task adaptation training schemes have proven their robustness in natural language processing across multiple deep learning architectures and prediction tasks [59, 60]. Taken together, these findings indicate benefits of this approach also for improving chemical compound transformer models such as ChemBERTa."

To clarify content relating to option (2), as suggested, we included the ChemBERTa model into the extensive state-of-the-art comparison of the manuscript (now including six different models in addition to ChemLM). Notably, we did not create further ChemBERTa-derived models using alternative training schemes, but benchmarked versus the existing model described in the original publications and typically assessed by others (e.g. MolFormer) in that way as well. A comparison versus new ChemBERTa model trained with our training schemes would be sensible, if no other evidence for the general relevance of the ChemLM training scheme was provided, which, however, is not the case (see above). Additionally, we added the following to the results section:

p. 9, line 299: *"Of the available language processing models, we compared ChemLM to MolFormer[55], MolBERT[56] and ChemBERTa[57], the currently leading models in the field. For a comparison with ChemBERTa, we utilized the "PubChem10M_SMILES_BPE_180k" model from Huggingface, pretrained on ten million SMILES derived from the PubChem database."*

page 9, line 299: *"In addition, ChemBERTa that achieved a median F1 score of 0.33 across folds (Table A6, Fig. 3b), faced challenges particularly for the positive class, recording an F1-score of only 0.17 for that class in the 4th fold (Supplementary Table A7)."*

page 11, line 337: *"ChemBERTa had an F1-macro averaged score of 0.9 and 0.87 for the ClinTox and BBBP datasets, respectively (Tables 1 and Supplementary Table A9, Fig. 4), performing slightly less well than ChemLM (0.92 and 0.88, respectively). On the BACE*

dataset, ChemBERTa's performance was notably lower, achieving a macro-averaged F1 score of 0.69, compared to 0.8 for ChemLM (Supplementary Table A10)."

Table 1 Comparison of ChemLM on ClinTox dataset with its simpler versions, and state-of-the-art models in more evaluation metrics. The macro-averaged score is reported for each metric.

Model	F1	AUC	Precision	Recall	Accuracy
MolFormer	0.55	0.7	0.89	0.7	0.95
MolBERT	0.67	0.67	0.66	0.67	0.92
ChemBERTa	0.9	0.84	0.99	0.84	0.98
MPNN	0.64	0.59	0.87	0.59	0.94
GAT	0.59	0.57	0.72	0.57	0.93
GCNN	0.66	0.61	0.78	0.61	0.94
ChemLM vanilla	0.48	0.50	0.46	0.50	0.93
ChemLM domain-adapted	0.82	0.75	0.98	0.75	0.96
ChemLM domain-adapted & optimized	0.92	0.86	0.99	0.86	0.98

Table A6 Performance comparison of property prediction models over the test set of a 5-fold cross-validation setting over the experimental dataset. The macro-averaged score is reported for each metric.

Model	F1	AUC	Precision	Recall	Accuracy
MolFormer	0.45	0.49	0.47	0.47	0.73
MolBERT	0.5	0.5	0.55	0.6	0.71
ChemBERTa	0.33	0.5	0.25	0.5	0.5
MPNN	0.60	0.59	0.66	0.59	0.79
GAT	0.56	0.58	0.64	0.58	0.71
GCNN	0.57	0.57	0.58	0.57	0.65
ChemLM	0.90	0.90	0.90	0.90	0.90

The median metric value of each model is demonstrated.

Table A7 Predictive performance of *ChemLM* and state-of-the-art models on the positive class (highly potent pathoblockers).

Hierarchical Folds	ChemLM	MPNN	GAT	GCNN	MolBERT	MolFormer	ChemBERTa
1	0.92	0.94	0.92	0.91	0.87	0.45	0.47
2	0.83	0.8	0.82	0.77	0.78	0.45	0.4
3	0.90	0.72	0.62	0.63	0.70	0.73	0.29
4	0.89	0.33	0.00	0.75	0.00	0.42	0.17
5	0.89	0.75	0.73	0.00	0.60	0.67	0.33

The F1-score is reported as evaluation metric in this this table.

Table A9 Comparison of *ChemLM* on BBBP dataset with its simpler versions, and state-of-the-art models in more evaluation metrics. The macro-averaged score is reported for each metric.

Model	F1	AUC	Precision	Recall	Accuracy
MolFormer	0.92	0.71	0.89	0.71	0.86
MolBERT	0.89	0.89	0.9	0.89	0.93
ChemBERTa	0.87	0.89	0.85	0.89	0.9
MPNN	0.78	0.79	0.78	0.79	0.84
GAT	0.75	0.71	0.85	0.71	0.85
GCNN	0.7	0.66	0.82	0.66	0.83
ChemLM vanilla	0.69	0.67	0.72	0.67	0.8
ChemLM domain-adapted	0.82	0.81	0.837	0.81	0.87
ChemLM domain-adapted & optimized	0.88	0.89	0.87	0.88	0.91

Table A10 Comparison of *ChemLM* on BACE dataset with its simpler versions, and state-of-the-art models in more evaluation metrics. The macro-averaged score is reported for each metric.

Model	F1	AUC	Precision	Recall	Accuracy
MolFormer	0.9	0.91	0.91	0.91	0.91
MolBERT	0.81	0.81	0.81	0.81	0.82
ChemBERTa	0.69	0.69	0.69	0.69	0.69
MPNN	0.73	0.73	0.73	0.73	0.73
GAT	0.67	0.70	0.77	0.70	0.68
GCNN	0.69	0.69	0.73	0.69	0.71
ChemLM vanilla	0.51	0.55	0.60	0.55	0.58
ChemLM domain-adapted	0.66	0.66	0.67	0.66	0.67
ChemLM domain-adapted & optimized	0.80	0.80	0.80	0.80	0.81

Table A11 Comparison of *ChemLM* on benchmark datasets with state-of-the-art models in prediction of the positive class using F1-score as evaluation metric.

Model	ClinTox	BACE	BBBP
MolFormer	0.55	0.90	0.92
MolBERT	0.38	0.8	0.96
ChemBERTa	0.81	0.67	0.93
MPNN	0.31	0.72	0.90
GAT	0.22	0.73	0.91
GCNN	0.35	0.61	0.90
ChemLM domain-adapted & optimized	0.84	0.79	0.94

b

Fig. 3 Description of experimental data: (a) Chemical structures and number of compounds per class. (b) Performance comparison of *ChemLM* with graph neural networks and transformers - based approaches in 5-fold validation for experimental compounds on *Pseudomonas aeruginosa*. The graph neural networks (blue) are GAT (Graph Attention Transformers) [16], MPNN (Message Passing Neural Networks)[14], and GCNN (Graph Convolutional Neural Networks)[54]. MolBERT [55], MolFormer [56] and ChemBERTa [34] are transformer-based approaches. *ChemLM* model is noted in red. Gray dots represent the F1 scores achieved by the model across the five folds.

Fig. 4 Performance of *ChemLM* and state-of-the-art models with the macro averaged F1-score on the test data sets of the benchmark data. *ChemLM* and its variations are compared with state-of-the-art models. Red diamonds represent the mean macro-averaged F1-score for each model across the three datasets.

Minor Comments

1. Please check the abbreviations, such as "masked language modeling." It should be "MLM," as the authors defined it as MLM when it first appeared.

Thank you for pointing this out. We have modified the manuscript accordingly.